# Maternal mortality associated with COVID-19 in Brazil in 2020 and 2021: Comparison with non-pregnant women and men

**Beatriz Martinelli Menezes Gonçalves**[1]*, **Rossana Pulcinelli V. Franco**[1], **Agatha S. Rodrigues**[2]

**1** Disciplina de Obstetrícia, Departamento de Obstetrícia e Ginecologia, Faculdade de Medicina da Universidade de São Paulo, São Paulo, São Paulo, Brasil, **2** Departamento de Estatística, Universidade Federal do Espírito Santo, Vitória, Espírito Santo, Brasil

* beatrizmartinellimg@gmail.com

**Data Availability Statement:** The data that support the findings of this study are available in GitHub repository at https://github.com/observatorioobstetrico/COVID19_2020vs2021.

## Abstract

### Objective

Mortality rates of pregnant and postpartum women grew in the second COVID-19 pandemic year. Our objective is to understand this phenomenon to avoid further deaths.

### Methods

We collected data from SIVEP-Gripe, a nationwide Brazilian database containing surveillance data on all severe acute respiratory syndrome caused by COVID-19, between the first notified case (February 2020) until the 17th epidemiological week of 2021. We stratified patients into maternal women (which includes pregnant and postpartum women), non-maternal women and men and divided them by time of diagnosis in two periods: first period (February to December 2020) and second period (the first 17 epidemiological weeks of 2021 before pregnant and postpartum women were vaccinated).

### Results

During the second period, all patients had higher risk of presenting severe COVID-19 cases, but the maternal population was at a higher risk of death (OR of 2.60 CI 95%: 2.28–2.97)– almost double the risk of the two other groups. Maternal women also had a higher risk of needing intensive care, intubation and of presenting desaturation in the second period. Importantly, maternal women presented fewer comorbidities than other patient groups, suggesting that pregnancy and postpartum can be an important risk factor associated with severe COVID-19.

### Conclusion

Our results suggest that the Gama variant, which has been related to greater virulence, transmissibility and mortality rates leads to more severe cases of COVID-19 for pregnant and postpartum women.

These data were derived from the following resources available in the public domain: https://opendatasus.saude.gov.br/dataset/bd-srag-2020 and https://opendatasus.saude.gov.br/dataset/bd-srag-2021 obtained on May 6, 2021. The first period (8th to 53rd epidemiological week of 2020) and the second period (1st to 17th epidemiological week of 2021) datasets can be obtained at https://kaggle.com/agatharodrigues/papermaternal.

**Funding:** This work was supported, in whole or in part, by the Bill & Melinda Gates Foundation (Seattle, USA - INV-027961). Under the grant conditions of the Foundation, a Creative Commons Attribution 4.0 Generic License has already been assigned to the Author Accepted Manuscript version that might arise from this submission. This work is also funded by Conselho Nacional de Desenvolvimento Científico e Tecnológico (CNPq - Brasília, Brazil - Award Number: 445881/2020-8) and Fundação de Amparo à Pesquisa e inovação do Estado do Espírito Santo (FAPES - Espírito Santo, Brazil - Award Number: 007/2021). The funders had no role in study design, data collection and analysis, decision to publish, or preparation of the manuscript.

**Competing interests:** The authors have declared that no competing interests exist.

## Introduction

The new coronavirus has been so far responsible for about 3.52 million deaths around the world and about 600 thousand deaths in Brazil, inflicting a great demand and pressure on the health system [1]. Initially, it had been thought that COVID-19 did not have a higher virulence in obstetric patients [2–5], however, recent studies have highlighted an elevated risk of complications in pregnant and postpartum women, with greater demand for admission into intensive care unit and need for mechanical ventilation [6–12]. The maternal mortality rate has had an increase of about 20% in Brazil during 2020, according to recent data [13, 14]. In early 2021, the maternal mortality rate seems to be even higher, even though the knowledge about the disease and the treatment tools have evolved. A recent study by Takemoto et al., demonstrated that the mortality rate was about two times higher for the maternal population in 2021 (15.6%), when compared to 2020 (7.4%) [15, 16].

The second pandemic year has been associated with the discovery of new SARS-CoV-2 variants, such as the Gama variant, which has been related to greater virulence, transmissibility and mortality rates [17–19]. The variant first emerged in Manaus, Brazil, in November 2020, initially with a low prevalence. It had increased to about 73% of the isolated strains by January 2021 [19, 20]. In São Paulo, one of the country's most populated cities, Gama was found in about 77% of patients with COVID-19 in November 2020 [21, 22]. The rapid dissemination of the Gama variant, exponentially increasing the number of COVID-19 related hospital admissions, weakened an already fragile health system [21]. The increase in maternal deaths in early 2021 seems to be related to an increasing prevalence of the Gama SARS-CoV-2 variant [16].

The rise in the maternal mortality rate agrees with the escalation of deaths among non-pregnant women and men during the first weeks of 2021, early in the second pandemic year [21, 22]. Knowing that maternal deaths have increased in Brazil during 2021, we found it important to compare both periods (2020 and the first 17 epidemiological weeks of 2021) and different patient groups (maternal women, non-maternal women and men) to evaluate symptoms, access to healthcare and comorbidities to understand if this increase in mortality rate in maternal women is significant when compared to the other population groups.

## Materials and methods

For our analysis, we used data available at the SIVEP-Gripe (Sistema de Informação de Vigilância Epidemiológica da Gripe), which is a nationwide record database for all notified cases of severe acute respiratory syndrome (SARS). In Brazil, all cases of SARS are notified, as both public and private hospitals are obliged to report them.

The database contains information regarding demographic characteristics (gender, age, race, gestational status, location, and education), clinical features including symptoms and comorbidities, laboratory diagnosis confirming the cause of SARS, as well as information regarding need for critical care, intubation, and date of death, if appliable. Specifically, cases of SARS or death caused by SARS regardless of hospitalization were then further filtered into confirmed cases of COVID-19. These were selected to be part of the study only if they had an enclosed outcome of death or cure, and as such were further stratified by epidemiological week. The search was limited to the first notified case of COVID-19 in February 2020 until the 17h epidemiological week of 2021 (up to May 1, 2021), the datasets were obtained on May 6, 2021. For the analysis, only variables correctly and completely informed were accounted. Three groups of patients were created: maternal women (composed of pregnant women of any gestational age and women until the 42nd day of postpartum), non-maternal women, and men. No age limit was established for the non-maternal women and men populations, although a limit of 10 to 55 years was determined for the maternal population. Cases in which

infection by COVID-19 were not confirmed by a positive lab test, or no data entry regarding pregnancy or postpartum status were available, or outcome of cure or death were not documented were excluded. The final result was 975,109 total cases, further divided into epidemiological week and year of COVID-19 cases occurring in the first period, from February to December 2020 (n= 603,313), or second period, from January to May 1 2021, the first 17 epidemiological weeks of 2021 (n=371,796); and stratified into three subgroups of men, non-maternal women and maternal women according to the date of diagnosis (Fig 1). This database is managed by the Health Ministry of Brazil (Ministério da Saúde and the Secretaria de Vigilância cia em Saúde).

We analyzed epidemiological characteristics (age and age group, sex, number of comorbidities, schooling, self-reported skin color), clinical outcomes (death or cure, necessity of ICU

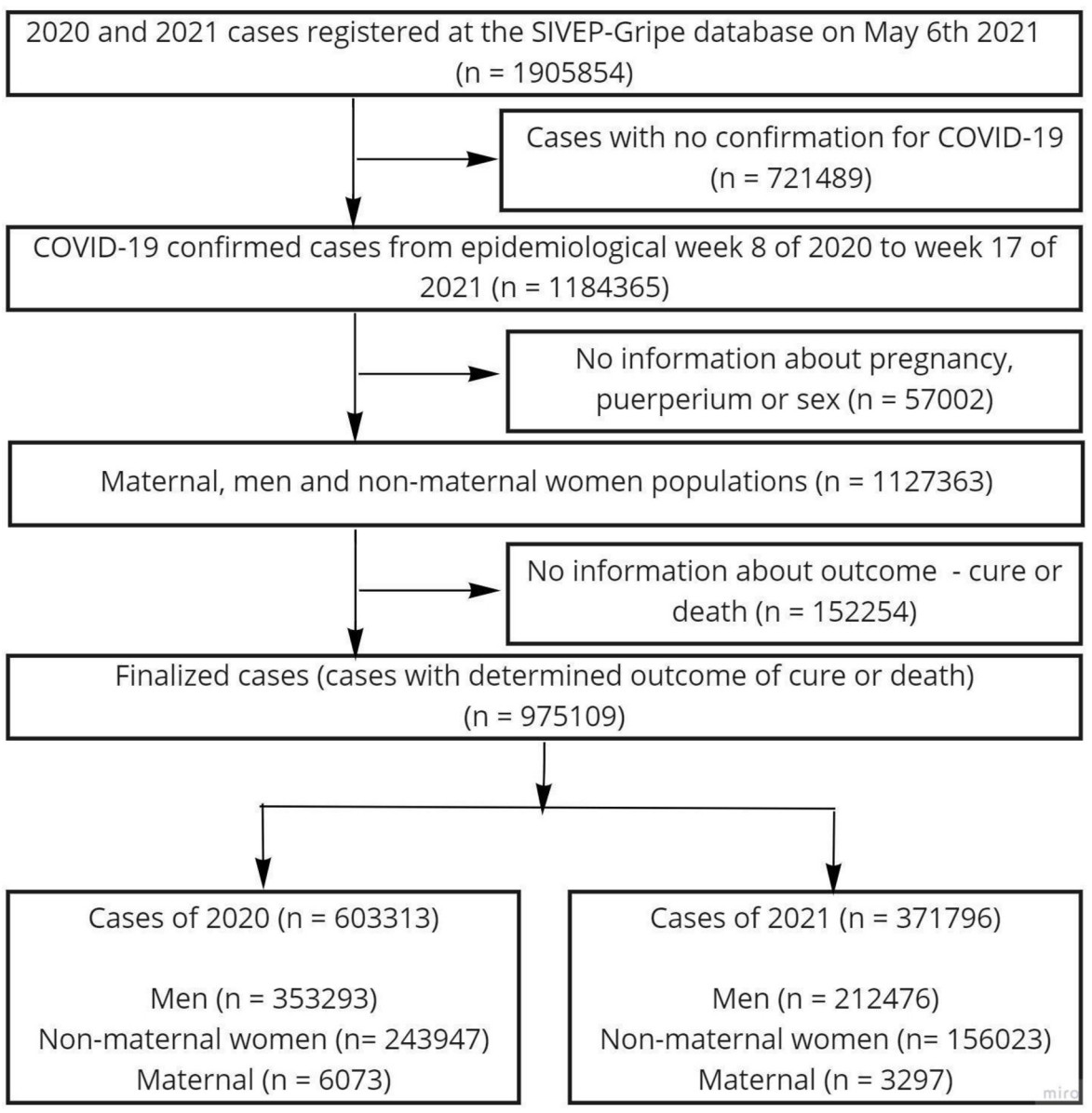

**Fig 1. Case selection and stratification.**

treatment, intubation and respiratory support), symptoms (cough, fever, dyspnea, fatigue, respiratory discomfort, desaturation, sore throat, diarrhea, vomit, abdominal pain, ageusia and dysgeusia, as well as the presence of any symptom or any respiratory symptom) and comorbidities (diabetes, obesity, asthma, cardiac, renal, hepatic, hematologic, neurologic and pneumological comorbidities, as well as any comorbidity). The analysis was performed for the three subgroups regarding the first and second periods taking into consideration the outcome, recovery or death.

## Data analysis

Quantitative variables were summarized as mean and standard deviation. Qualitative variables were displayed as absolute frequencies (n) and category percentages (%).

T-test was applied to compare the two periods in each study group in terms of quantitative variables. Chi-square test was used to evaluate the association between groups and qualitative variables. Odds Ratio (OR) was considered as a measure of association to compare the relative odds of the occurrence of the outcome of interest between first and second periods with a 95% confidence interval (CI 95%). The forest plot was used to visualize the OR of pandemic years comparison in each of the three groups. In order to compare the OR for pandemic years comparison among the three groups, the Breslow Day test was applied [23].

The significance level adopted is 5%. The exception will be for Breslow Day tests, in which we have considered the Bonferroni correction [24]. The overall significance level is 5% and so, the adjusted significance level is 0.05/3=0.0167, since we have three 2-to-2 comparisons.

The analyses were performed with the statistical R software (R Foundation for Statistical Computing Platform, version 4.0.3) [25].

SIVEP-Gripe is an open database, with no possibility of individual identification and, therefore, according to Brazilian regulations, this study does not require prior approval by the institutional ethics board review.

## Results

A total of 975,109 cases with confirmed diagnosis of COVID-19 were analyzed divided into two periods. The first period accounts for cases from February to December 2020; the second period accounts for cases from January to May 1, 2021, which corresponds to the first 17 epidemiological weeks of 2021, according to the date of confirmed diagnosis. In the first period, 603,313 COVID-19 cases were stratified into three subgroups of non-maternal women (n = 243,947; 61% of all COVID-19 cases in non-maternal women in Brazil in 2020-2021), men (n = 353,293; 62.4% of all COVID-19 cases in men in Brazil in 2020-2021) and maternal patients (n = 6073; 64.8% of all COVID-19 cases in maternal women in Brazil in 2020-2021). The same stratification was reproduced for the 371,796 cases in the second period, which were divided into non-maternal women (n = 156,023; 39.0% of all COVID-19 cases in non-maternal women in Brazil in 2020-2021), men (n = 212,476; 37.6% of all COVID-19 cases in men in Brazil in 2020-2021) and maternal patients (n = 3,297; 35.2% of all COVID-19 cases in maternal women in Brazil in 2020-2021). The studied population regarding both pandemic periods was different when analyzing race, and education (Table 1). Non-maternal women were at a mean age of 61.67 years old (SD 19.16) in the first period and 61.47 years old (SD 17.21) in the second period. For men, the mean age was 58.98 years old (SD 17.99) in the first period and 58.05 years old (SD 16.89) in the second period. Among the maternal population, the mean age was 31.78 years old (SD 7.59) in the first period and 32.66 years old (SD 7.47) in the second period (Table 1).

**Table 1. Epidemiologic characteristics.**

| Epidemiologic characteristics | Year | Women | | Men | | Maternal | |
|---|---|---|---|---|---|---|---|
| | | n (%) or mean± SD | p value | n (%) or mean± SD | p value | n (%) or mean± SD | p value |
| Ethnicity | First period (February to December, 2020) | Yellow 2554/196987 (1.3%) | < 0.001a | Yellow 3958/277184 (1.4%) | < 0.001a | Yellow 56/5009 (1.1%) | < 0.001a |
| | | White 100109/196987 (50.8%) | | White 134864/277184 (48.7%) | | White 1765/5009 (35.2%) | |
| | | Indigenous 697/196987 (0.4%) | | Indigenous 1120/277184 (0.4%) | | Indigenous 84/5009 (1.7%) | |
| | | Brown 82234/196987 (41.7%) | | Brown 120967/277184 (43.6%) | | Brown 2788/5009 (55.7%) | |
| | | Black 11393/196987 (5.8%) | | Black 16275/277184 (5.9%) | | Black 316/5009 (6.3%) | |
| | Second period (January to May 1, 2021) | Yellow 1293/135121 (1.0%) | | Yellow 1841/177476 (1.0%) | | Yellow 20/2810 (0.7%) | |
| | | White 75534/135121 (55.9%) | | White 96392/177476 (54.3%) | | White 1252/2810 (44.6%) | |
| | | Indigenous 226/135121 (0.2%) | | Indigenous 317/177476 (0.2%) | | Indigenous 14/2810 (0.5%) | |
| | | Brown 51691/135121 (38.3%) | | Brown 70260/177476 (39.6%) | | Brown 1352/2810 (48.1%) | |
| | | Black 6377/135121 (4.7%) | | Black 8666/177476 (4.9%) | | Black 488/2810 (6.2%) | |
| Education | First period | Fundamental 1 26962/92198 (29.2%) | < 0.001a | Fundamental 1 33236/129345 (25.7%) | < 0.001a | Fundamental 1 232/2719 (8.5%) | < 0.001a |
| | | Fundamental 2 16892/92198 (18.3%) | | Fundamental 2 23895/129345 (18.5%) | | Fundamental 2 514/2719 (18.9%) | |
| | | Median 27248/92198 (29.6%) | | Median 41934/129345 (32.4%) | | Median 1453/2719 (53.4%) | |
| | | No school 8193/92198 (8.9%) | | No school 8119/129345 (6.3%) | | No school 17/2719 (0.6%) | |
| | | Superior 12903/92198 (14.0%) | | Superior 22161/129345 (17.1%) | | Superior 503/2719 (18.5%) | |
| | Second period | Fundamental 1 18657/58579 (31.8%) | | Fundamental 1 20697/100240 (26.5%) | | Fundamental 1 107/1444 (5.4%) | |
| | | Fundamental 2 11071/58579 (18.9%) | | Fundamental 2 14639/100240 (18.7%) | | Fundamental 2 281/1444 (19.5%) | |
| | | Median 16675/58579 (28.5%) | | Median 25731/100240 (33.0%) | | Median 767/1444 (53.1%) | |
| | | No school 4601/58579 (7.9%) | | No school 4280/100240 (5.5%) | | No school 10/1444 (0.7%) | |
| | | Superior 7575/58579 (12.9%) | | Superior 34893/100240 (16.8%) | | Superior 279/1444 (19.3%) | |
| Age | First period | 61.67 ± 19.16 | < 0.001b | 58.98 ± 17.99 | < 0.001b | 29.64 ± 7.57 | < 0.001b |
| | Second period | 61.47 ± 17.21 | | 58.05 ± 16.89 | | 30.85 ± 7.46 | |
| Age group | First period | < 20 years 5885/243947 (2.4%) | < 0.001a | < 20 years 7120/353290 (2.0%) | < 0.001a | < 20 years 552/6073 (9.1%) | < 0.001 |
| | | 20–34 years 15781/243947 (6.5%) | | 20–34 years 23825/353290 (6.7%) | | 20–34 years 3925/6073 (64.6%) | |
| | | ≥ 35 years 222281/243947 (91.1%) | | ≥ 35 years 322345/353290 (91.2%) | | ≥ 35 years 1596/6073 (26.3%) | |
| | Second period | < 20 years 2206/156020 (1.4%) | | < 20 years 2571/212473 (1.2%) | | < 20 years 173/3297 (5.2%) | |
| | | 20–34 years 8023/156020 (5.1%) | | 20–34 years 14685/212473 (6.9%) | | 20–34 years 2091/3297 (63.4%) | |
| | | ≥ 35 years 145791/156020 (93.4%) | | ≥ 35 years 195217/212473 (91.9%) | | ≥ 35 years 1033/3297 (31.3%) | |

Number of patients (n) and percentage (%), or mean ± standard deviation (SD) for each specific category. Proportions in pandemic periods were compared using a Chi-square test (superscript a). Comparison between pandemic periods for mean age was performed using t-test (superscript b).

Education level was divided as Fundamental 1 (5 school years), Fundamental 2 (9 school years), Median (12 school years), No school and Superior (University degree, at least 15 school years).

**Table 2. Comorbidities.**

| Comorbidities | Year | Women | | Men | | Maternal | |
|---|---|---|---|---|---|---|---|
| | | n (%) | p value a | n (%) | p value a | n (%) | p value a |
| Any comorbidity | First period (February to December, 2020) | 141625/155956 (90.8%) | 0.296 | 184567/203790 (90.6%) | <0.001 | 1226/2712 (45.2%) | 0.005 |
| | Second period (January to May 1, 2021) | 87680/96685 (90.7%) | | 102645/113857 (90.2%) | | 696/1397 (49.8%) | |
| Cardiac | First period | 91019/137072 (66.4%) | 0.587 | 116104/177605 (65.4%) | 0.45 | 405/2436 (16.6%) | 0.5205 |
| | Second period | 56260/84582 (66.5%) | | 64404/98734 (65.2%) | | 212/1209 (17.5%) | |
| Hematologic | First period | 2064/103819 (2.0%) | < 0.001 | 2478/133971 (1.8%) | < 0.001 | 43/2331 (1.8%) | 0.899 |
| | Second period | 981/63967 (1.5%) | | 1081/74574 (1.4%) | | 23/1167 (2.0%) | |
| Hepatic | First period | 1740/103381 (1.7%) | < 0.001 | 3658/133883 (2.7%) | < 0.001 | 24/2308 (1.0%) | 0.40 |
| | Second period | 921/63751 (1.4%) | | 1604/74492 (2.2%) | | 8/1160 (0.7%) | |
| Asthma | First period | 8561/105581 (8.1%) | < 0.001 | 6891/135182 (5.1%) | 0.012 | 234/2380 (9.8%) | 0.99 |
| | Second period | 4739/65003 (7.3%) | | 3644/75150 (4.8%) | | 116/1189 (9.8%) | |
| Diabetes | First period | 68333/128694 (53.1%) | < 0.001 | 84109/165323 (50%) | < 0.001 | 427/2441 (17.5%) | 0.011 |
| | Second period | 40107/78789 (50.9%) | | 43529/91139 (47.8%) | | 261/1244 (21.0%) | |
| Neurologic | First period | 11445/107077 (10.7%) | < 0.001 | 12492/137136 (9.1%) | 0.84 | 40/2321 (1.7%) | 0.223 |
| | Second period | 5307/65262 (8.1%) | | 5753/75986 (7.6%) | | 13/1158 (1.1%) | |
| Pneumologic | First period | 9839/106340 (9.3%) | < 0.001 | 13605/137559 (9.9%) | < 0.001 | 41/2318 (1.8%) | 0.90 |
| | Second period | 4500/65153 (6.9%) | | 5842/76105 (7.7%) | | 22/1162 (1.9%) | |
| Renal | First period | 9039/105630 (8.6%) | < 0.001 | 15580/138023 (11.3%) | < 0.001 | 49/2311 (2.1%) | 0.90 |
| | Second period | 4194/64655 (6.5%) | | 6603/76207 (8.7%) | | 23/1155 (2.0%) | |
| Obesity | First period | 17095/105908 (16.1%) | < 0.001 | 19291/136516 (14.1%) | < 0.001 | 264/2355 (11.2%) | < 0.001 |
| | Second period | 16567/68375 (24.2%) | | 17110/79051 (21.6%) | | 235/1219 (19.3%) | |

Number of patients (n) and percentage (%) for each specific comorbidity and comparison between pandemic periods for non-maternal women, men and maternal women. Comparison between pandemic period was performed using a chi-square test.

Almost 90% of men and non-maternal women presented at least one comorbidity. In the maternal group, at least one comorbidity was only observed in less than 50% of the cases. A greater prevalence of obesity was observed within all three groups in the second period when compared to the first one (Table 2). During the first 17 weeks of the second pandemic year, the prevalence of comorbidities in men dropped (Table 2). In non-maternal women, the only comorbidity that was similar to that in the first period was neurologic disease (Table 2). In the maternal population, there was a higher risk of diabetes, occurring together with any other comorbidity in the first 17 weeks of 2021, while values observed in both periods for the other comorbidities were similar.

In the first 17 weeks of 2021, patients were at higher risk of presenting symptoms associated with severe cases of COVID-19 such as dyspnea and desaturation (Table 3, Fig 2). For the maternal population, the risk of desaturation in the second period was 1.52 times higher than in the first period (OR 2.52 CI 95%: 2.29-2.78) (Table 3, Fig 2). Also in the second period, all three groups were at a higher risk of presenting any respiratory symptom: for non-maternal women the risk was 0.74 times higher (OR 1.74 CI 95%: 1.69-1.78), for men 0.63 times higher (OR 1.63 CI 95%: 1.60-1.67) and for the maternal population 1.02 times higher (OR 2.02 CI 95%: 1.81-2.26) (Table 3, Fig 2). There was also an increased risk of developing fatigue within all three groups, with a more prominent risk among the maternal population (OR 2.05 CI 95%: 1.81-2.33) (Table 3, Fig 2). The forest graph shows the comparative risk ratio for all three

**Table 3. Symptoms reported by patients.**

| Symptoms | Year | Women | | Men | | Maternal | | Breslow-Day - OR comparison | | |
|---|---|---|---|---|---|---|---|---|---|---|
| | | n (%) | OR (95% CI) | n (%) | OR (95% CI) | n (%) | OR (95% CI) | Maternal vs Women | Maternal vs Men | Women vs Men |
| Any symptom | First period (February to December, 2020) | 233905/ 237299 (98.6%) | 1.51 (1.42– 1.61) | 339154/ 343473 (98.7%) | 1.43 (1.36– 1.51) | 5430/5879 (92.4%) | 1.88 (1.54– 2.29) | 0.402 | 0.009 | 0.193 |
| | Second period (January to May 1, 2021) | 151028/ 152477 (99.0%) | | 205057/ 206881 (99.1%) | | 3065/3200 (95.8%) | | | | |
| Any respiratory symptoms | First period | 207555/ 228281 (90.9%) | 1.74 (1.69- 1.78) | 301008/ 329571 (91.3%) | 1.63 (1.60– 167) | 3813/5491 (69.4%) | 2.02 (1.81– 2.26) | 0.007 | < 0.001 | < 0.001 |
| | Second period | 1397535/ 147777 (94.6%) | | 189235/ 200237 (94.5%) | | 2470/3007 (82.1%) | | | | |
| Fever | First period | 138543/ 209334 (66.2%) | 0.84 (0.83– 0.86) | 226803/ 308362 (73.6%) | 0.80 (0.79– 0.81) | 3403/5376 (63.3%) | 0.98 (0.89– 1.07) | 0.003 | < 0.001 | < 0.001 |
| | Second period | 80249/128853 (62.3%) | | 122752/ 178177 (68.9%) | | 1741/2776 (62.7%) | | | | |
| Cough | First period | 166697/ 215598 (77.3%) | 0.99 (0.97– 1.00) | 248973/ 312856 (79.6%) | 0.93 (0.92– 0.95) | 3949/5481 (72.0%) | 1.18 (1.07- 1.31) | < 0.001 | < 0.001 | < 0.001 |
| | Second period | 103934/ 134840 (77.1%) | | 143355/ 182714 (78.5%) | | 2204/2927 (75.3%) | | | | |
| Sore throat | First period | 44588/ 180093 (24.8%) | 1.01 (1.00– 1.03) | 61288/ 256602 (23.9%) | 1.02 (1.01– 1.04) | 1305/4828 (27.0%) | 1.06 (0.95– 1.19) | 0.400 | 0.510 | 0.377 |
| | Second period | 27429/109630 (25.0%) | | 35734/ 146877 (24.3%) | | 686/2428 (28.3%) | | | | |
| Dyspnea | First period | 169052/ 216344 (78.1%) | 1.34 (1.32– 1.36) | 245924/ 311947 (78.8%) | 1.29 (1.28– 1.31) | 2952/5268 (56.0%) | 1.78 (1.62– 1.96) | < 0.001 | < 0.001 | 0.003 |
| | Second period | 114424/ 138310 (82.7%) | | 155289/ 187499 (82.8%) | | 1984/2857 (69.4%) | | | | |
| Respiratory discomfort | First period | 137170/ 201792 (68.0%) | 1.24 (1.22– 1.26) | 198115/ 289968 (68.3%) | 1.22 (1.20– 1.24) | 2417/5058 (47.8%) | 1.52 (1.38– 1.67) | < 0.001 | < 0.001 | 0.163 |
| | Second period | 92738/128050 (72.4%) | | 125306/ 172943 (72.5%) | | 1544/2654 (58.2%) | | | | |
| Desaturation | First period | 140133/ 204453 (68.5%) | 1.80 (1.77– 1.82) | 205273/ 294073 (69.8%) | 1.72 (1.70– 1.74) | 1581/4943 (32.0%) | 2.52 (2.29– 2.78) | < 0.001 | < 0.001 | < 0.001 |
| | Second period | 106303/ 133477 (79.6%) | | 144085/ 180321 (79.9%) | | 1452/2676 (54.3%) | | | | |
| Diarrhea | First period | 36947/178466 (20.7%) | 1.04 (1.02– 1.06) | 44238/ 252705 (17.5%) | 1.07 (1.06– 1.09) | 644/4694 (13.7%) | 1.18 (1.03– 1.36) | 0.075 | 0.178 | 0.015 |
| | Second period | 23266/108928 (21.4%) | | 26881/ 144942 (18.5%) | | 373/2360 (15.8%) | | | | |

(*Continued*)

**Table 3.** (Continued)

| Symptoms | Year | Women | | Men | | Maternal | | Breslow-Day - OR comparison | | |
|---|---|---|---|---|---|---|---|---|---|---|
| | | n (%) | OR (95% CI) | n (%) | OR (95% CI) | n (%) | OR (95% CI) | Maternal vs Women | Maternal vs Men | Women vs Men |
| Vomit | First period | 22918/174831 (13.1%) | 1.08 (1.06–1.10) | 23506/247372 (9.5%) | 1.08 (1.05–1.10) | 600/4673 (12.8%) | 1.06 (0.92–1.23) | 0.837 | 0.869 | 0.846 |
| | Second period | 14934/106570 (14.0%) | | 14368/141430 (10.2%) | | 317/2340 (13.5%) | | | | |
| Abdominal pain | First period | 9466/108507 (8.7%) | 1.13 (1.10–1.17) | 10965/151434 (7.2%) | 1.19 (1.16–1.23) | 254/2563 (9.9%) | 1.16 (0.96–1.39) | 0.822 | 0.731 | 0.008 |
| | Second period | 10118/103654 (9.85) | | 11826/138677 (8.5%) | | 256/2269 (11.3%) | | | | |
| Fatigue | First period | 33533/112324 (29.9%) | 1.55 (1.52–1.58) | 45628/157278 (29.0%) | 1.55 (1.53–1.57) | 543/2594 (20.9%) | 2.05 (1.81–2.33) | < 0.001 | < 0.001 | 0.974 |
| | Second period | 44042/110878 (39.7%) | | 57574/148529 (38.8%) | | 843/2395 (35.2%) | | | | |
| Olfactory loss | First period | 16743/109439 (15.3%) | 1.03 (1.01–1.05) | 22185/153103 (14.5%) | 1.05 (1.03–1.07) | 673/2670 (25.2%) | 0.92 (0.81–1.05) | 0.094 | 0.050 | 0.247 |
| | Second period | 16495/105146 (15.7%) | | 21251/140781 (15.1%) | | 560/2364 (23.7%) | | | | |
| Loss of taste | First period | 16838/109120 (15.4%) | 1.04 (1.01–1.06) | 22262/152615 (14.6%) | 1.07 (1.05–1.09) | 589/2631 (22.4%) | 1.01 (0.88–1.15) | 0.709 | 0.434 | 0.075 |
| | Second period | 16712/105024 (15.9%) | | 21667/140616 (15.4%) | | 533/2361 (22.6%) | | | | |

Number of patients (n) and percentage (%) for each specific symptom as well as odds ratio with 95% confidence interval (OR (IC 95%)) and statistical comparison between two pandemic periods for non-maternal women, men and maternal women. p-value represent Breslow Day test comparison between OR.

groups regarding each symptom, thus highlighting that the severe respiratory symptoms associated with COVID-19 were higher for the maternal population in the first weeks of 2021 (Fig 2).

The maternal population had a higher risk of death due to COVID-19 in the first 17 epidemiological weeks of 2021 (OR of 2.60 CI 95%: 2.28–2.97) (Table 4, Fig 3). Although the risk of dying was also greater among non-maternal women (OR of 1.44 CI 95%: 1.42-1.46) and men (OR of 1.31 CI 95%: 1.30-1.32) when comparing both periods, it was at least three times higher for pregnant and postpartum women (Table 4, Fig 3). The maternal population was also exposed to a higher risk of needing support in critical care (OR 1.60 CI 95%: 1.45–1.77) and intubation (OR 2.02 CI 95%: 1.78–2.30). Meanwhile men were at a lower risk of needing admission into critical care during the second period (OR 0.98 CI 0.97-0.99) (Table 4, Fig 3). The maternal population was at a higher risk for all three outcomes when compared to men and women in the second period (Table 4, Fig 3).

To conclude, maternal patients were at a significantly higher risk of suffering from severe symptoms and cases of COVID-19 of requiring ICU admission and intubation and of dying of

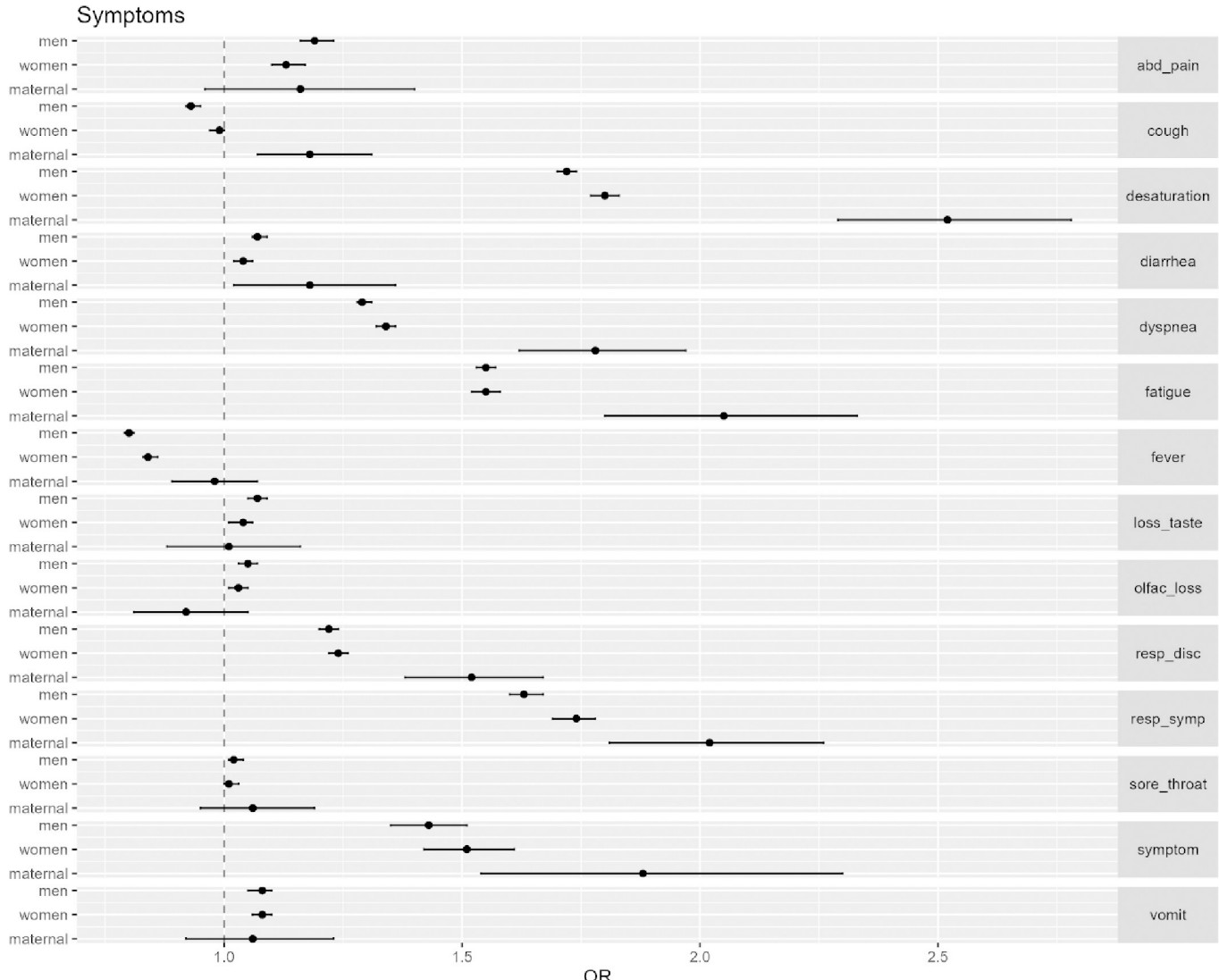

**Fig 2. Symptoms forest plot.** Maternal women are at higher risk of severe COVID-19 symptoms. Odds ratios (OR) for individual symptoms or group of symptoms for men, non-maternal women (women) and maternal women (maternal). *(Definitions: abd pain – abdominal pain; loss taste – ageusia; olfac loss – olfactory loss (anosmia); resp disc – respiratory discomfort; symptom – any symptom).*

COVID19 in the first 17 epidemiological weeks of 2021, if compared to non-maternal women and men.

## Discussion

During the first 17 epidemiological weeks of 2021, maternal patients were more exposed to unsatisfactory outcomes and severe clinical presentations related to COVID-19, if compared to other populations. Hence, pregnant and postpartum women were at a higher risk of dying than the rest of the population. This finding can be associated with the elevated risk of presenting any symptom, as well as SARS related symptoms (such as desaturation and dyspnea) viciously more prevalent for the maternal population. The early weeks of the year 2021, the beginning of the second pandemic year, also exposed the maternal population to a greater risk of intubation and admission into ICU, two times higher than the one for men and women

**Table 4. Clinical outcomes.**

| Outcomes | Year | Women | | Men | | Maternal | | OR comparison | | |
|---|---|---|---|---|---|---|---|---|---|---|
| | | n (%) | OR (95% CI) | n (%) | OR (95% CI) | n (%) | OR (95% CI) | Maternal vs Women | Maternal vs Men | Women vs Men |
| Intubation | First period (February to December, 2020) | 40791/ 206707 (19.7%) | 1.37 (1.35–1.39) | 64019/296497 (21.6%) | 1.23 (1.22–1.25) | 531/5303 (10.0%) | 2.02 (1.78–2.30) | < 0.001 | < 0.001 | < 0.001 |
| | Second period (January to May 1, 2021) | 34304/ 136294 (25.2%) | | 46591/183827 (25.3%) | | 547/2981 (18.3%) | | | | |
| ICU | First period | 76281/ 208612 (36.6%) | 1.02 (1.01–1.03) | 119690/ 302498 (39.6%) | 0.98 (0.97–0.99) | 1245/5388 (23.1%) | 1.60 (1.45–1.77) | < 0.001 | < 0.001 | < 0.001 |
| | Second period | 50789/ 137210 (37.0%) | | 72831/186251 (39.1%) | | 981/3019 (32.5%) | | | | |
| Death | First period | 85356/ 243947 (35.0%) | 1.44 (1.42–1.46) | 127671/ 353293 (36.1%) | 1.31 (1.30–1.32) | 456/6073 (7.5%) | 2.60 (2.28–2.97) | < 0.001 | < 0.001 | < 0.001 |
| | Second period | 68101/ 156023 (43.6%) | | 90451/212476 (42.6%) | | 575/3197 (17.4%) | | | | |

Number of patients (n) and percentage (%) for each specific outcome as well as odds ratio with 95% confidence interval (OR (IC 95%)) and statistical comparison between two pandemic periods for non-maternal women, men and maternal women. p-value represent Breslow Day test comparison between OR.

during the same period. These results outline the fragility of the maternal group to respond to COVID-19 during the first weeks of 2021.

Pregnant and postpartum women have been shown to be at a higher risk of developing more severe illnesses related to the SARS-COV-2 infection as highlighted by Zambrano et al. in November 2020 [11, 26]. This data was further corroborated by a systematic review published by Wastnedge et al., also suggesting a higher vulnerability of the pregnant population to the infection [27]. One of the possibilities to justify the worst prognosis of pregnant women is postpartum is recent studies suggest that the expression of angiotensin converting enzyme 2 (ACE 2), a receptor used by SARS-CoV-2 in human placentas, could increase maternal population susceptibility to infection [28]. ACE 2 receptors actively participate in necessary hemodynamic adaptations through pregnancy, when consumed by SARS-CoV-2 this could increase risk for placental dysfunction [29, 30].

Patients infected by SARS-CoV-2 were at higher risk of presenting severe symptoms and needing invasive ventilatory support during the first weeks of 2021. Although these results were observed for non-maternal women and for men, the risk was extremely higher among the maternal population. Studies also validateours, suggesting that pregnant women are at a higher risk for ICU care during the 2021 pandemic [3, 4]. The elevated risk of death, intubation, and need of ICU care for the maternal population during 2021 are corroborated by similar literature [6, 13, 16].

Genome isolation was not done thus we cannot say what cases were associated with which SARS-CoV-2 variant. However, it is known that during 2021 the P1 SARS-CoV-2 variant was the one most prevalent in Brazil [16–20]. Therefore, our findings might be explained by a higher mortality rate associated with new coronavirus variants as seen in other studies such as Ellington et al., (OR for admission into ICU of 1.5 CI 1.2-1.8), Zambrano and colleagues (increased risk of admission into ICU OR 6.6 IC 4.0-11.0), and Sentilhes, et al., who observed that 23% of the studied pregnant population needed invasive respiratory support [8, 11, 26].

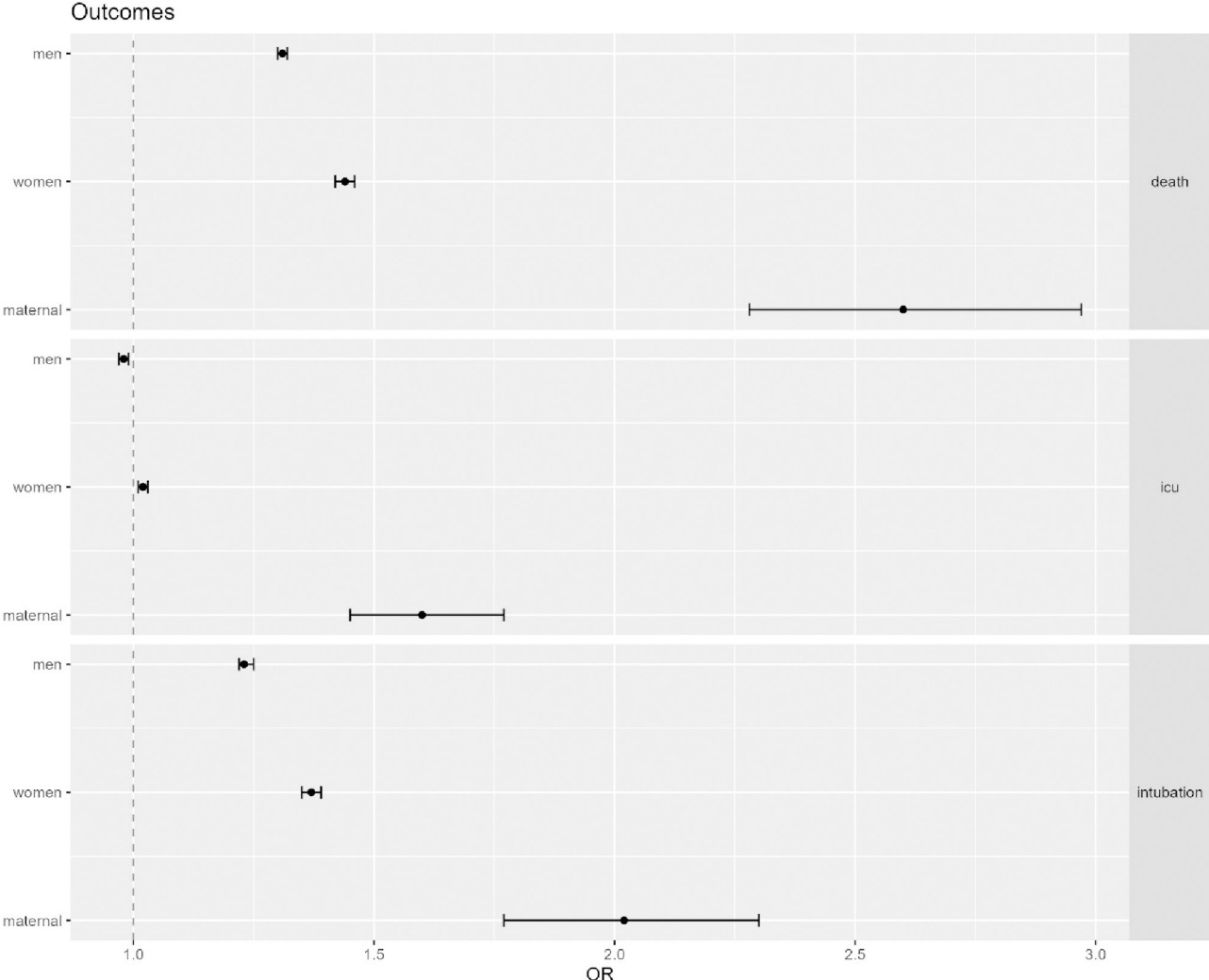

**Fig 3. Outcomes forest plot.** Maternal women are at higher risk of negative clinical outcomes of COVID-19. Odds ratios (OR) for clinical outcomes for men, non-maternal women (women) and maternal women (maternal): death; intubation; uci – ICU admission.

Additionally, all groups were at a higher risk of obesity in the first 17 epidemiological weeks of 2021, a known risk factor associated with unsuccessful COVID-19 related outcomes. Thus, it can corroborate the increase in death risk observed for all three groups during the second pandemic year [3025]. Yet, for the maternal group, less than 50% of the cases presented at least one comorbidity, while approximately 90% men and non-maternal women presented comorbidities. Considering that the analyzed database only has cases of SARS-CoV-2 infection, this conclusion outlines what has been highlighted by previous studies: pregnancy and postpartum alone are an important risk factor associated with worst outcomes associated to COVID-19 [6–12].

Documenting that pregnant and postpartum women are at a higher risk of developing severe forms of COVID-19 is essential for prioritizing them in health care strategies. Considering the already fragile health systems and scarce access to resources, this conclusion should influence public health decisions regarding vaccination priorities, which is still ongoing in Brazil. Knowing that this population endures a higher risk for death, intubation and ICU admission, health care workers should closely monitor all pregnant and postpartum women infected

by COVID-19 or be even more permissive with criteria for hospital admission. Analyzing pregnant women infected with SARS-CoV-2 that are evolving into severe clinical presentations with fetal compromise, can also be an important criteria for births. Understanding why the maternal population is at a higher risk of presenting worst outcomes, specifically in relation to their immunologic response could also be a glimpse into new treatments or medications against COVID-19.

The period studied is prior to the start of vaccination of the maternal population. Vaccination in Brazil for the maternal population began in May 2021. Our study confirms the worse evolution of the population of pregnant and postpartum women when compared to non-pregnant women and men and may be useful in guiding this population regarding the importance of vaccination against COVID-19. Most references point out that the probability of complications arising during pregnancy due to SARS-COV2 infection is greater than those related to vaccination [31, 32]. Currently, the maternal and postpartum population is considered at high risk for SARS-COV2 infection and should be among the priorities for vaccination.

The large sample size and the use of a nationwide database are one of our study's strengths, which allows analyzing different regions in a country with continental dimensions as Brazil, most of which presenting social and economic disparities. Being able to compare the maternal population to other subgroups allowed us to observe that the effect of COVID-19 related mortality on pregnant and postpartum women was not equally present in the rest of the population during the 2021 pandemic year. This conclusion is different from other studies which analyze the maternal group independently [12–24]. Therefore, our study is unique in comparing symptoms, comorbidities, and outcomes between the maternal patients and the rest of the population and between the different pandemic years [16].

The large sample size can also be considered a limitation, it can induce statistically significant results, sometimes with no clinical applicability or difference. This can be exemplified by the differences noted in the population characteristics such as ethnicity and education, which hardly differ in percentage but have a significant p value. Probably the difference in ethnicity and education among both pandemic years is consequence of the sample size. Another limitation is the amount of missing data and possible errors of data entry within the database. As a way to minimize error, which is commonly found in population databases, the variables were only analyzed if complete information was available.

The presence of health disorders and the role of physical activity were not analyzed in the present study. Increased mortality in patients with mental disorders has been described by Fond et al. in previous studies, OR (1.38 [95% CI, 1.15-1.65]; P < .05). [33] Physical activity was also identified as a protective factor, both for aerobic and muscular strengthening activities (0.02% vs 0.08%; aRR 0.24; 95% CI 0.05 to 0.99) [34].

Lastly, genome sequencing of the viruses causing these cases was not performed. As such, we could not evaluate which cases were caused by new SARS-CoV-2 variants. However, it is known that during 2021 the P1 SARS-CoV-2 variant was the most one prevalent in Brazil [16–20].

It seems that the increased death in the beginning of the second pandemic year in Brazil is associated with the prevalence of the new COVID-19 variants. Also, the maternal population seems to be more vulnerable to those new variants. Yet, this conclusion cannot yet be definitively established. An analysis of the COVID-19 genome, and a prevalence study of the types of COVID-19 variants infecting the maternal population in comparison to the other groups could aid in that understanding. Another hypothesis for increased virulence and mortality in the maternal population could possibly be a particular immune response to the virus, which could be less effective in suppressing the new variants in this population.

Our results suggest a higher severity with the new variant for pregnant and postpartum women, even though that effect was also observed for men and women during 2021. Until the entire population is vaccinated and a deeper understanding about COVID-19 is generated, there will be no end to the coronavirus crisis in the world.

## Author Contributions

**Conceptualization:** Beatriz Martinelli Menezes Gonçalves, Rossana Pulcinelli V. Franco, Agatha S. Rodrigues.

**Data curation:** Beatriz Martinelli Menezes Gonçalves, Rossana Pulcinelli V. Franco, Agatha S. Rodrigues.

**Formal analysis:** Beatriz Martinelli Menezes Gonçalves, Rossana Pulcinelli V. Franco, Agatha S. Rodrigues.

**Funding acquisition:** Agatha S. Rodrigues.

**Investigation:** Beatriz Martinelli Menezes Gonçalves, Rossana Pulcinelli V. Franco, Agatha S. Rodrigues.

**Methodology:** Beatriz Martinelli Menezes Gonçalves, Rossana Pulcinelli V. Franco, Agatha S. Rodrigues.

**Project administration:** Beatriz Martinelli Menezes Gonçalves, Rossana Pulcinelli V. Franco.

**Resources:** Beatriz Martinelli Menezes Gonçalves.

**Software:** Agatha S. Rodrigues.

**Supervision:** Rossana Pulcinelli V. Franco.

**Validation:** Beatriz Martinelli Menezes Gonçalves, Rossana Pulcinelli V. Franco, Agatha S. Rodrigues.

**Visualization:** Beatriz Martinelli Menezes Gonçalves, Rossana Pulcinelli V. Franco, Agatha S. Rodrigues.

**Writing – original draft:** Beatriz Martinelli Menezes Gonçalves, Rossana Pulcinelli V. Franco.

**Writing – review & editing:** Beatriz Martinelli Menezes Gonçalves, Rossana Pulcinelli V. Franco, Agatha S. Rodrigues.

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
