## [Decision Letter · Decision Letter 0]

18 Oct 2021

PONE-D-21-29410Maternal mortality associated with COVID-19 in Brazil in 2020 and 2021: comparison with non-pregnant women and menPLOS ONE

Dear Dr. Gonçalves,

Thank you for submitting your manuscript to PLOS ONE. After careful consideration, we feel that it has merit but does not fully meet PLOS ONE’s publication criteria as it currently stands. Therefore, we invite you to submit a revised version of the manuscript that addresses the points raised during the review process.

We look forward to receiving your revised manuscript.

Kind regards,

Dong Keon Yon, MD, FACAAI

Academic Editor

PLOS ONE

Journal Requirements:

"This work was supported, in whole or in part, by the Bill & Melinda Gates Foundation [INV-027961]. Under the grant conditions of the Foundation, a Creative Commons Attribution 4.0 Generic License has already been assigned to the Author Accepted Manuscript version that might arise from this submission."

"YES - Bill & Melinda Gates Foundation (Seattle, USA), Conselho Nacional de Desenvolvimento Científico e Tecnológico (CNPQ - Brasília, Brazil), and Fundação de amparo à pesquisa e inovação do Estado do Espírito Santo (FAPES - Espírito Santo, Brazil) are funding source. Their role is to support the research."

"No author have. competing interests"

Additional Editor Comments:

I congratulate you on your mesmerizing work. I have raised several concerns.

#1. Documenting that pregnant and postpartum women are at a higher risk of developing severe forms of COVID-19 isessential for prioritizing them in health care strategies.

Please cite references.

#2. Please explain plausible basic mechanism. (i.e., ACE2, SARS-CoV-2, pregnancy, molecular mechanism)

#3. The authors have to provide the policy implication regarding vaccination. Please see the landmark previos study.

Hwang J, Park SH, Lee SW, Lee SB, Lee MH, Jeong GH, Kim MS, Kim JY, Koyanagi A, Jacob L, Jung SY, Song J, Yon DK, Shin JI, Smith L. Predictors of mortality in thrombotic thrombocytopenia after adenoviral COVID-19 vaccination: the FAPIC score. Eur Heart J. 2021 Sep 21:ehab592. doi: 10.1093/eurheartj/ehab592. Epub ahead of print. PMID: 34545400; PMCID: PMC8500026.

#4. Limitation

Pregnancy patients was associted with depression [1] and low level physical activity [2], which may mediated the COVID-19 effect.

Please disscuss this in the limitation section and cite the top-tier papers below.

[1] Fond G, Nemani K, Etchecopar-Etchart D, Loundou A, Goff DC, Lee SW, Lancon C, Auquier P, Baumstarck K, Llorca PM, Yon DK, Boyer L. Association Between Mental Health Disorders and Mortality Among Patients With COVID-19 in 7 Countries: A Systematic Review and Meta-analysis. JAMA Psychiatry. 2021 Jul 27:e212274. doi: 10.1001/jamapsychiatry.2021.2274. Epub ahead of print. PMID: 34313711; PMCID: PMC8317055.

[2] Lee SW, Lee J, Moon SY, Jin HY, Yang JM, Ogino S, Song M, Hong SH, Abou Ghayda R, Kronbichler A, Koyanagi A, Jacob L, Dragioti E, Smith L, Giovannucci E, Lee IM, Lee DH, Lee KH, Shin YH, Kim SY, Kim MS, Won HH, Ekelund U, Shin JI, Yon DK. Physical activity and the risk of SARS-CoV-2 infection, severe COVID-19 illness and COVID-19 related mortality in South Korea: a nationwide cohort study. Br J Sports Med. 2021 Jul 22:bjsports-2021-104203. doi: 10.1136/bjsports-2021-104203. Epub ahead of print. PMID: 34301715; PMCID: PMC8300550.

Again, I congratulate your excellent paper.

Reviewers' comments:

Reviewer's Responses to Questions

**Comments to the Author**

1. Is the manuscript technically sound, and do the data support the conclusions?

Reviewer #1: Yes

Reviewer #2: Yes

2. Has the statistical analysis been performed appropriately and rigorously? 

Reviewer #1: Yes

Reviewer #2: Yes

3. Have the authors made all data underlying the findings in their manuscript fully available?

Reviewer #1: Yes

Reviewer #2: Yes

4. Is the manuscript presented in an intelligible fashion and written in standard English?

Reviewer #1: Yes

Reviewer #2: Yes

5. Review Comments to the Author

Reviewer #1: A very valuable article in my opinion, this article can be published as a scientific reference for many related research, the method of posing the problem was effective and linking the problem with the results was strong, the presentation of the information was consistent and understandable, the statistical method is strong and convincing, my only comment, I may need more introduction to understand the problem In the region of Brazil. I highly recommend this search

Reviewer #2: Good use of the routine data to answer important program question. Covid 19 pandemic has impacted public health globally. National level technical committees are faced with the questions to identify high risk groups in order to priortise public health interventions. Women with pregnancy were not included for covid vaccination due to lack of data and evidence. Later countries included this group based on the clinical evidence. Researchers in this manuscript have created evidence that pregnancy is a high risk group. This is important despite all its limitations.

6. PLOS authors have the option to publish the peer review history of their article (what does this mean?). If published, this will include your full peer review and any attached files.

Reviewer #1: **Yes: **Issam J. A Abu Qeis

Reviewer #2: **Yes: **Dr Arun Kumar Aggarwal

---

## [Author Response · Author response to Decision Letter 0]

30 Nov 2021

1. #1. Documenting that pregnant and postpartum women are at a higher risk of developing severe forms of COVID-19 is essential for prioritizing them in health care strategies.

Please cite references.

Response: Thank you for your comment. The following text has been included in the current version of the manuscript in the discussion section. 

“Pregnant and postpartum women have been shown to be at a higher risk of developing more severe illnesses related to the SARS-COV-2 infection as highlighted by Zambrano et al. in November 2020 [11]. This data was further corroborated by a systematic review published by Wastnedge et al, also suggesting a higher vulnerability of the pregnant population to the infection [27]. Other studies also validateours, suggesting that pregnant women are at a higher risk for ICU care during the 2021 pandemic. [13,16].”

References: 

Zambrano LD, Ellington S, Strid P, Galang RR, Oduyebo T, Tong VT, et al. Update: Characteristics of Symptomatic Women of Reproductive Age with Laboratory-Confirmed SARS-CoV-2 Infection by Pregnancy Status - United States, January 22-October 3, 2020. MMWR Morb Mortal Wkly Rep. 2020;69(44):1641-1647.

Wastnedge EAN, Reynolds RM, van Boeckel SR, Stock SJ, Denison FC, Maybin JA, Critchley HOD. Pregnancy and COVID-19. Physiol Rev. 2021 Jan 1;101(1):303-318. doi: 10.1152/physrev.00024.2020. Epub 2020 Sep 24. PMID: 32969772; PMCID: PMC7686875.

Nakamura-Pereira M, Andreucci CB, Menezes M., Knobel R. et al. Worldwide maternal deaths due to COVID-19: A brief review. Gynecology & Obstetrics. 2020; 152: 148-150.

Takemoto MLS, Menezes M de O de O, Andreucci CB, et al. Clinical characteristics and risk factors for mortality in obstetric patients with severe COVID-19 in Brazil: a surveillance database analysis. BJOG 2020; 127: 1618–26.

2. Please explain plausible basic mechanism. (i.e., ACE2, SARS-CoV-2, pregnancy, molecular mechanism)

Response:Thank you for your suggestion. The following text has been included in the current version of the manuscript. 

Recent studies suggest that the expression of angiotensin converting enzyme 2 (ACE 2), a receptor used by SARS-CoV-2 in human placentas, could increase maternal population susceptibility to infection. [1] ACE 2 receptors actively participate in necessary hemodynamic adaptations through pregnancy, when consumed by SARS-CoV-2 this could increase risk for placental dysfunction increasing risk of worst outcomes after infection [29,30].

References :

29. Nobrega Cruz N. A., Stoll D., Casarini D. E., Bertagnolli M. Role of ACE2 in pregnancy and potential implications for COVID-19 susceptibility. ClinSci (Lond) 13 August 2021; 135 (15): 1805–1824. doi: https://doi.org/10.1042/CS20210284

30. Todros T, Masturzo B, De Francia S. COVID-19 infection: ACE2, pregnancy and preeclampsia. Eur J ObstetGynecolReprod Biol. 2020;253:330. doi:10.1016/j.ejogrb.2020.08.007

3. The authors have to provide the policy implication regarding vaccination. Please see the landmark previos study. 

Hwang J, Park SH, Lee SW, Lee SB, Lee MH, Jeong GH, Kim MS, Kim JY, Koyanagi A, Jacob L, Jung SY, Song J, Yon DK, Shin JI, Smith L. Predictors of mortality in thrombotic thrombocytopenia after adenoviral COVID-19 vaccination: the FAPIC score. Eur Heart J. 2021 Sep 21:ehab592. doi: 10.1093/eurheartj/ehab592. Epub ahead of print. PMID: 34545400; PMCID: PMC8500026.

Thank you for your comment. We did not provide any information regarding vaccination policy for pregnant and postpartum women because the studied period is previous to the maternal population vaccination start. The vaccination in Brazil for maternal population started in May 2021.

The following text has been included in the current version of the manuscript (Discussion): 

“The period studied is prior to the start of vaccination of the maternal population. Vaccination in Brazil for the maternal population began in May 2021. Our study confirms the worse evolution of the population of pregnant and postpartum women when compared to non-pregnant women and men and may be useful in guiding this population regarding the importance of vaccination against COVID-19. Most references point out that the probability of complications arising during pregnancy due to SARS-COV2 infection is greater than those related to vaccination [31,32]. Currently, the maternal and postpartum population is considered at high risk for SARS-COV2 infection and should be among the priorities for vaccination.”

31. Rasmussen et al. COVID-19 Vaccines and Pregnancy. American College of Obstetricians and Gynecologists. 2021; 137 (3).

32. Russell, F. M. and Greenwood, B. Who should be prioritised for COVID-19 vaccination? Human Vaccin Immunother. 2021; 17(3): 1317-1321. 

4. Pregnancy patients was associted with depression [1] and low level physical activity [2], which may mediated the COVID-19 effect.

Please disscuss this in the limitation section and cite the top-tier papers below.

Thank you for your suggestion. The following remarks were added to our discussion into the limitation section.

“The presence of health disorders and the role of physical activity were not analyzed in the present study. Increased mortality in patients with mental disorders has been described by Fond et al. in previous studies, OR (1.38 [95% CI, 1.15-1.65]; P < .05). [33] Physical activity was also identified as a protective factor, both for aerobic and muscular strengthening activities (0.02% vs 0.08%; aRR 0.24; 95% CI 0.05 to 0.99) [34]”

33. Fond G, Nemani K, Etchecopar-Etchart D, Loundou A, Goff DC, Lee SW, Lancon C, Auquier P, Baumstarck K, Llorca PM, Yon DK, Boyer L. Association Between Mental Health Disorders and Mortality Among Patients With COVID-19 in 7 Countries: A Systematic Review and Meta-analysis. JAMA Psychiatry. 2021 Jul 27:e212274. doi: 10.1001/jamapsychiatry.2021.2274. Epub ahead of print. PMID: 34313711; PMCID: PMC8317055.

34. Lee SW, Lee J, Moon SY, Jin HY, Yang JM, Ogino S, Song M, Hong SH, AbouGhayda R, Kronbichler A, Koyanagi A, Jacob L, Dragioti E, Smith L, Giovannucci E, Lee IM, Lee DH, Lee KH, Shin YH, Kim SY, Kim MS, Won HH, Ekelund U, Shin JI, Yon DK. Physical activity and the risk of SARS-CoV-2 infection, severe COVID-19 illness and COVID-19 related mortality in South Korea: a nationwide cohort study. Br J Sports Med. 2021 Jul 22:bjsports-2021-104203. doi: 10.1136/bjsports-2021-104203. Epub ahead of print. PMID: 34301715; PMCID: PMC8300550.

Reviewer #2: 

 I may need more introduction to understand the problem in the region of Brazil.

Thank you for your comment. The following text was added to our paper. 

“In early 2021, the maternal mortality rate seems to be even higher, even though the knowledge about the disease and the treatment tools have evolved. A recent study by Takemoto et. al, demonstrated that the mortality rate was about two times higher for the maternal population in 2021 (15.6%), when compared to 2020 (7.4%), but these authors did not compare the group of pregnant and postpartum women with the groups of men and women , as proposed in this study (7.4%) [15,16]. “

15. Takemoto MLS, Menezes M de O de O, Andreucci CB, et al. Higher case fatality rate among obstetric patients with COVID-19 in the second year of pandemic in Brazil: do new genetic variants play a role? medRxiv doi:https://doi.org/10.1101/2021.05.06.21256651.

16. Takemoto MLS, Menezes M de O de O, Andreucci CB, et al. Clinical characteristics and risk factors for mortality in obstetric patients with severe COVID-19 in Brazil: a surveillance database analysis. BJOG 2020; 127: 1618–26.

---

## [Editor Report · Decision Letter 1]

3 Dec 2021

Maternal mortality associated with COVID-19 in Brazil in 2020 and 2021: comparison with non-pregnant women and men

PONE-D-21-29410R1

Dear Dr. Gonçalves,

We’re pleased to inform you that your manuscript has been judged scientifically suitable for publication and will be formally accepted for publication once it meets all outstanding technical requirements.

Kind regards,

Dong Keon Yon, MD, FACAAI

Academic Editor

PLOS ONE

Additional Editor Comments (optional):

I congratulate you on your mesmerizing work.
---

## [Editor Report · Acceptance letter]

10 Dec 2021

PONE-D-21-29410R1 

Maternal mortality associated with COVID-19 in Brazil in 2020 and 2021: comparison with non-pregnant women and men 

Dear Dr. Gonçalves:

I'm pleased to inform you that your manuscript has been deemed suitable for publication in PLOS ONE. Congratulations! Your manuscript is now with our production department. 

Kind regards, 

on behalf of

Dr. Dong Keon Yon 

Academic Editor

PLOS ONE